# Foliar Application of Low Concentrations of Titanium Dioxide and Zinc Oxide Nanoparticles to the Common Sunflower under Field Conditions

**DOI:** 10.3390/nano10081619

**Published:** 2020-08-18

**Authors:** Marek Kolenčík, Dávid Ernst, Martin Urík, Ľuba Ďurišová, Marek Bujdoš, Martin Šebesta, Edmud Dobročka, Samuel Kšiňan, Ramakanth Illa, Yu Qian, Huan Feng, Ivan Černý, Veronika Holišová, Gabriela Kratošová

**Affiliations:** 1Department of Soil Science and Geology, Faculty of Agrobiology and Food Resources, Slovak University of Agriculture in Nitra, Tr. A. Hlinku 2, 949 76 Nitra, Slovakia; 2Nanotechnology Centre, VŠB Technical University of Ostrava, 17. listopadu 15/2172, 708 00 Ostrava-Poruba, Czech Republic; Gabriela.Kratosova@vsb.cz; 3Department of Crop Production and Grassland Ecosystems, Faculty of Agrobiology and Food Resources, Slovak University of Agriculture in Nitra, Tr. A. Hlinku 2, 949 76 Nitra, Slovakia; david.ernst@uniag.sk (D.E.); ivan.cerny@uniag.sk (I.Č.); 4Institute of Laboratory Research on Geomaterials, Faculty of Natural Sciences, Comenius University in Bratislava, Mlynska dolina, Ilkovičova 6, 842 15 Bratislava, Slovakia; martin.urik@uniba.sk (M.U.); marek.bujdos@uniba.sk (M.B.); martin.sebesta@uniba.sk (M.Š.); 5Department of Environment and Biology, Faculty of Agrobiology and Food Resources, Slovak University of Agriculture in Nitra, Tr. A. Hlinku 2, 949 76 Nitra, Slovakia; luba.durisova@uniag.sk (Ľ.Ď.); xksinans@is.uniag.sk (S.K.); 6Institute of Electrical Engineering, Slovak Academy of Sciences, Dúbravská cesta 9, 841 04 Bratislava, Slovakia; edmund.dobrocka@savba.sk; 7Department of Chemistry, Rajiv Gandhi University of Knowledge Technologies, AP IIIT, Nuzvid 521202, Krishna District, India; ramakanthilla@yahoo.com; 8School of Ecology and Environmental Science, Yunnan University, 2 Cuihubei Lu, Kunming 650091, China; qianyu@ynu.edu.cn; 9Department of Earth and Environmental Studies, Montclair State University, 1 Normal Ave, Montclair, NJ 070 43, USA; fengh@montclair.edu; 10Laboratory of Growth Regulators, Faculty of Science, Palacký University, Šlechtitelů 27, 7837 Olomouc, Czech Republic; HolisovaVeronika@seznam.cz

**Keywords:** foliar application, titanium dioxide, zinc oxide, nanoparticles, sunflower, nano-fertilisers

## Abstract

Nano-fertilisers have only recently been introduced to intensify plant production, and there still remains inadequate scientific knowledge on their plant-related effects. This paper therefore compares the effects of two nano-fertilisers on common sunflower production under field conditions. The benefits arising from the foliar application of micronutrient-based zinc oxide fertiliser were compared with those from the titanium dioxide plant-growth enhancer. Both the zinc oxide (ZnO) and titanium dioxide (TiO_2_) were delivered by foliar application in nano-size at a concentration of 2.6 mg·L^−1^. The foliar-applied nanoparticles (NPs) had good crystallinity and a mean size distribution under 30 nm. There were significant differences between these two experimental treatments in the leaf surfaces’ trichomes diversity, ratio, width, and length at the flower-bud development stage. Somewhat surprisingly, our results established that the ZnO-NPs treatment induced generally better sunflower physiological responses, while the TiO_2_-NPs primarily affected quantitative and nutritional parameters such as oil content and changed sunflower physiology to early maturation. There were no differences detected in titanium or zinc translocation or accumulation in the fully ripe sunflower seeds compared to the experimental controls, and our positive results therefore encourage further nano-fertiliser research.

## 1. Introduction

Nanoparticles (NPs) are defined as chemical entities with at least one of their three dimensions less than 100 nm and have significantly different physical, chemical and biological properties to their macro-sized and dissolved ionic counterparts [1,2,3,4]. The NPs and nanomaterials have unique high surface energy and specific surface area and quantum size effects. This enables their frequent application in optical [5] and medical devices [6], electronics [7], pharmaceutics, and biotechnology [8,9,10].

Nanoparticles are also extremely useful in agriculture where they can alleviate the effects of plant diseases and are active components in nano-fertilisers [11,12,13,14,15,16,17,18]. Liu and Lal [15] classify nano-fertilisers according to the following categories: (i) macro-nutrient nano-fertilisers; (ii) micro-nutrient nano-fertilisers that include nanoparticles which are mainly oxides such as ZnO, CuO, Fe_2_O_3_, etc.; (iii) nutrient-augmented nanomaterials, such as zeolites, and (iv) plant-growth enhancers with unpredictable action, such as TiO_2_ and carbon nanotubes.

Nano-fertilisers have effective plant penetration by foliar-application or through the root system [13,19] and the agricultural foliar application of NPs is widely recognised as an effective tool for growth enhancement in the common sunflower (*Helianthus annuus* L.). Research reports indicate that the sunflower disposes of broad leaves [20] and it has appropriate stomata morphology [21] and also the ability to absorb, differentiate and accumulate metals in several parts of the plant. This is especially evident in the leaf structures [2,22,23,24,25]. The common sunflower is part of the Asteraceae family. It is one of the most attractive world oil-bearing crops [26,27], and its by-products have been successfully incorporated in a wide spectrum of applications, including phytoremediation [28], green-fertilisation [29], animal feeding, and bio-fuels [30,31].

Zinc is an essential plant micro-nutrient, and it is usually supplemented into the fertilisers in the form of solid ZnO [24,32,33,34]. Zinc is a cofactor for over three hundred enzymes and has important functions in gene expression and protein synthesis, and it also structurally and functionally stabilises membranes and proteins [4,16,35]. ZnO occurs in nature as a zincite mineral with wurtzite crystal symmetry [36]. It is part of the II–VI semiconductor group and is frequently used in ceramics, glass, pigments, and food additives [37].

The high electron mobility and wide band energy make ZnO a strong photocatalyst [37,38]. Furthermore, ZnO-NPs are reported to have antibacterial properties [1]. Hussain et al. [33] also indicate that ZnO-NPs positively affect plant growth and grain yields, and Torabian et al. [24] report that it helps ameliorate salt stress in sunflower cultivars such as “Olsion”. Finally, our previous study [39] indicated that ZnO-NPs foliar application resulted in higher foxtail millet oil content than ZnO-NPs-free treatment.

Although ionic titanium’s function in plant cells is not fully understood, low-level concentrations improve plant physiology and stress response [40], and Khater [41] recorded increased coriander yields after foliar TiO_2_-NP application. The TiO_2_-NPs also act as functional photocatalyst, and encourage photosynthesis and plant growth at certain wavelengths [42,43,44,45,46]. While the overall TiO_2_-NPs effects reported for plant growth are often confusing and contradictory [19,41,42,47,48], the inconsistences most likely reflect different experimental design. These include TiO_2_-NPs concentration, morphology and size distribution, their application method, and the target plant species’ uptake mechanisms [3,19,45,46].

Most referenced papers on the interaction of plants and nanoparticles are laboratory or green-house studies, and they mainly target early plant growth development under regulated conditions [11,17,49,50]. Consequently, there are only rare long-term field studies on NPs ability to accumulate elements required for best fruit quality.

Our hypothesis therefore compares the effects of foliar application of micro-based ZnO-NPs fertiliser and combine two-polymorphs of TiO_2_-NPs plant-growth enhancer on the *Helianthus annuus* L. common sunflower over one season. The sunflower has great ability to orientate against excessive sunlight radiation which could cause photocatalysis and accelerate both types of nanoparticles as photoactive nano-domains against selective photosynthetic plants. We predicted the various sunflower responses based on the nanoparticles’ distinct photo-corrosive ability and subsequent assimilation by leaves with different bioavailability and functional metal pathways. This comparison includes the sunflower’s quantitative, nutritional, and physiological parameters and its leaf morphological properties and fruit metal content.

## 2. Materials and Methods

### 2.1. Characterisation of Titanium Dioxide and Zinc Oxide Nanoparticles

The titanium dioxide nanoparticles (TiO_2_-NPs) were purchased from Sigma-Aldrich (Saint-Louis, MO, USA) as nano-powder with ≥99.5% metal basis, and they were analysed by the Bruker D8 DISCOVER diffractometer for X-ray diffraction (XRD) (Bruker, MA, USA). This was equipped with a Cu anode operating at 300 mA, 40 kV, and 12 kW, and the unit cell parameters were then estimated by TOPAS 3.0 software (Burker, MA, USA). TiO_2_-NP size distribution and morphology was investigated by JSM-7610F Plus scanning electron microscopy (JEOL, Tokyo Japan) and ZnO-NPs detailed characteristics are as reported in [39].

### 2.2. Plant Material

The experimental SY Neostar hybrid of the *Helianthus annuus* L. common sunflower (Syngenta, Basel, Switzerland) is part of the two-line imidazolinone-resistant hybrid suitable for the ClearField Plus^®^ production system. This hybrid has short-to-medium height, medium-to-early development, wide adaptability to different growing conditions and no special agricultural requirements [51]. In addition, it has medium-size thousand seed weight (TSW), produces free fatty acids with 47% average oil content and is tolerable to *Plasmopara halstedii* and resistant to *Diaporte helianthi* and *Sclerotinia sclerotiorum.*

### 2.3. Site Description and Monitoring Climate-Seasonal Variation

Field experiments were performed at Dolná Malanta in Nitra in the Slovak republic (N 48°19′25.41″ E 18°09′2.87″). This is the official research field of the Slovak University of Agriculture in Nitra (SUA Nitra). It lies south of the Tribeč mountains, in the north-eastern part of the Podunajská nížina lowland at 250 m above sea level, and close to the Žitavská pahorkatina. The petrology comprises granitic rocks, Mesozoic carbonates, sediments with neogenic and quarter creations, and eluvial-deluvial sediments in its upper parts [52]. The soil texture is silt loam haplic Luvisol [53], and the locality has been a maize-crop region with intensive soil-cultivation. The experimental sunflower is cultivated there in 7-plot crop rotation.

Climate variability was monitored by the SUA Nitra meteorological station during the 2018 growing season, and this included changes in daily temperature, the number of daylight hours, and precipitation accumulation.

### 2.4. Field Experiments

The field experiment was conducted on a 60 m^2^ land parcel, with three replications of each trial [54]. Three experimental variants were randomly organized in perpendicularly selected blocks (Figure 1).

The experimental field was deeply ploughed by Zetor 6211 tractor (Zetor Tractors, a. s., Brno, Czech Republic), where winter wheat had previously been cultivated. The soil was analysed before sunflower planting as in Hrivňáková et al. [55], Wierzbowska et al. [56], and Kováčik et al. [57], and nitrogen-related soil species were determined colorimetrically as in Šimanský et al. [58]. The soil characteristics are presented in Table 1 and Table 2.

Combined-type fertilisers containing 15% nitrogen, 15% P_2_O_5_ and 15% K_2_O (ACHP Levice, a. s., Levice, Slovak Republic) were applied by tractor as a pre-sowing soil treatment using the Ferti fertiliser applicator (Agromehanica, Boljevac, Serbia). The fertilisers were applied in 200 kg·ha^−1^ concentration based on soil agrochemical analysis before planting. The experimental SY Neostar sunflower hybrid was sown in lines with 60 mm sowing depth, 220 mm seed distance and 700 mm inter-row spacing by the Monosem NG Plus 3 planter (Monosem, Largeasse, France) [59]. Then, four L·ha^−1^ of Wing^®^ P herbicide (BASF, Ludwigshafen am Rhein, Germany) was applied pre-emergently and 0.5 L.ha^−1^ Pictor^®^ fungicide (BASF, Ludwigshafen am Rhein, Germany) was used 55 days after planting. All replications, including controls, were treated with herbicide and fungicide by AGT 865T/S sprayer (Agromehanica, Boljevac, Serbia). We must fully respect the principles of the sunflower cultivation system to achieve the required yield quantity and quality. This is based on the application of fertilisers, herbicides and fungicides [20,59]. Growth stimulator or foliar fertiliser application can be considered a cultivation system superstructure which has a positive effect on production, nutrition and physiological parameters in a changing climate [60]. These effects were confirmed in our previous sunflower experiments [61].

Plants were sprayed with 2.6 mg·L^−1^ TiO_2_-NPs or 2.6 mg·L^−1^ ZnO-NPs dispersed by GAMMA 10 hand sprayer (Mythos Di Martino, Mussolente, Italy). This was performed on early and windless mornings, and until the leaves were completely wet. Foliar application was conducted on the 40th day when the sunflower reached the phenological growth phase of leaf development and again on the 80th day when there was stem elongation with flower-bud formation, as in Meier [62] (Figure 2). The NPs-free controls had only water spraying. Ensuring adequate nutrient supply before flower initiation increases the number of sunflower grains and root biomass. While late fertiliser application only partly modifies weight per grain and mainly affects plant protein concentration and decreases oil concentration, application in the early crop stages can stimulate lush biomass. However, it also affects leaf area index (LAI) after flowering by disease proliferation, or by excessive water consumption which limits normal provision during grain-filling in limited water input conditions [59]. Finally, common cultivation practice applies foliar fertilisers at the two sunflower growth stages 40 and 80 days after planting, and our methodology follows the foliar application for growth conducted by Ernst et al. [61].

### 2.5. Microscopic Investigation of Sunflower Leaf Surface after Foliar Application of Nanoparticles

Sunflower leaves were collected for surface morphology analysis after the second foliar NPs application at flower-bud formation. The morphological differences were categorised in the following three trichomes groups by scanning electron microscopic examination with Quanta 450 FEG (FEI, Hillsboro, OR, USA): non-glandular (NGTs), linear glandular (LGTs), and capitate glandular (CGTs), as in Aschenbrenner et al. [63]. The proportion of the different trichome types was analysed on an approximately 7,500,000 μm^2^ leaf area, and their width and length were measured by Olympus cellSens Standard 1.9 software (Olympus corporation, Germany). Resultant data was statistically analysed by Tukey HSD test at α = 0.01 significance in version 10 Statistica software (StatSoft, Inc., Tulsa, OK, USA) [64].

### 2.6. Analysis of Sunflower Quantitative and Nutritional Production Parameters, and Translocation of Zinc and Titanium into Sunflower Seeds

The sunflower plants were harvested by small-plot combine Claas Dominator 38 when they reached full seed maturity (CLAAS GmbH & Co. KGaA, Harsewinkel, Germany). The seed moisture was analysed by HE Lite (Pfeuffer GmbH, Kitzingen, Germany), and the seed yield harvest was re-calculated to tonnes per hectare (t·ha^−1^). The sunflower plants from two random rows in each treatment block were then harvested to determine the following parameters: a manual count of plants and heads, head diameter in mm (Texi 4007 laboratory equipment; Texi GmbH, Berlin, Germany), head weight and TSW (Kern PCB3500-2 lab scale; KERN & Sohn GmbH, Balingen, Germany) and Numirex seed count (MEZOS spol. s r.o., Hradec Králové, Czech Republic). Finally, the percentage of sunflower oil content was quantified using Soxshlet method [65].

The translocated and accumulated zinc and titanium in the kernels and hulls of fully ripe sunflower seeds were analysed by ICP-MS (Thermo Scientific iCap-Q, Bremen, Germany). We then followed the standard measurement preparation: 0.15–0.30 g seed samples were digested in PTFE pressure vessels in the Anton Paar Multiwave 3000 microwave with concentrated 4 mL HNO_3_ and 2 mL H_2_O_2_ at 60 barometric pressure. The total zinc and titanium concentrations were determined by ICP-MS in KED mode with helium gas, and the calibration solutions were prepared from MERCK CertiPUR ICP 1000 mg·L^−1^ single-element standard solutions (Darmstadt, Germany). Scandium and rhodium provided the internal standards.

### 2.7. Investigation of Sunflower Photochemical Reflectance Index, Normalised Difference Vegetation Index and Crop Water Stress Index Physiological Parameters

The photochemical reflectance index (PRI), and normalised difference vegetation index (NDVI) were quantified by PlantPen 200-U (Photon Systems Instruments, Brno, Czech Republic). This detects reflected radiations from the leaf surface at 531 and 570 nm for PRI and 660 and 740 nm for NDVI [66]. Equations (1) and (2) provide PRI and NDV indices calculation:PRI = (R_531_ − R_570_)/(R_531_ + R_570_)(1)
NDVI = (R_740_ − R_660_)/(R_740_ + R_660_)(2)
where R_531_, R_570_, R_660_, and R_740_ are reflected radiation intensities from the leaf surface at indicated wavelengths.

Non-destructive methods were ensured for all measurements, and these were performed from 11.00 a.m. to 1.00 p.m. on similar dates, plants and growth phases. Each PRI and NDVI repetition on ten different annual sunflower mature leaves was labelled and measured during the growing season. Here, we used at least ten perpendicularly-oriented leaf measurement points for each index in order to cover all the leaf heterogeneity described by Gamon et al. [66].

The crop water stress index (CWSI) was calculated according to Jones et al. [67]. Measurement of atmospheric moisture, leaf temperature, and dry and wet leaf surface were required for the CWSI index calculation, and these were provided by EasIR-4 thermo-camera (Bibus AG, Fehraltorf, Switzerland). The thermal images gave diagonal sunflower scanning from 2 m distance, 1.5 m height, and 20.6° × 15.5° auto-focus field of view.

### 2.8. Statistical Analysis

All statistical analyses for TiO_2_-NPs and ZnO-NPs variants and NPs-free controls were determined in Statistica 10 software (StatSoft, Inc., Tulsa, OK, USA) [64]. Prior to the evaluation of the multifactorial analysis of variance (ANOVA), the normality of experimental data was tested at *α* = 0.05 and *α* = 0.01 significance by the Student *t*-test, Shapiro–Wilk test for trials, and Fisher’s least significant difference (LSD).

## 3. Results and Discussion

### 3.1. Characterisation of Titanium Dioxide and Zinc Oxide Nanoparticles and Foliar Application Effects on Surface Leaf Trichomes

Scanning electron microscopy shows that the TiO_2_-NPs are predominantly spherical and prismatic crystals with bi-pyramidal terminations and rarely occur as tabular crystals. This morphology corresponds to the rutile and anatase TiO_2_ polymorphic modifications identified by XRD analysis (Figure 3a,b). The mean size of the anatase and rutile crystals is approximately 19.6 ± 0.2 and 30.0 ± 2.0 nm, respectively (Table 3). The X-ray diffraction analysis in Table 3 also revealed both rutile and anatase have tetragonal symmetry and different unit cell parameters and relative content. The ZnO-NPs predominantly have spherical shape with 17.3 nm mean crystal size and wurtzite-type structure [39].

The sunflower’s effective nutrient uptake, development, growth and metabolic functions depend on a combination of factors. This is common to all plants undergoing improvement by NPs foliar application, and the most important factors include the plant species and cultivar, ambient light and water conditions and the NPs particle size, stoichiometry, crystallinity and concentration [68,69]. The two most usual nanoparticle penetration sites are cuticular and stomatal [70], but only the stomatal pathway was available to our research Larue et al. [22] because the cuticular pathway requires less than 5 nm NP-size and we were limited to 10 nm–1μm NP-size with relatively high transport velocity [70]. More precisely, both our nanoparticle types are less than 30 nm, and they have good stoichiometry and high crystallinity with typical morphology (Table 3, Figure 3). However, because of their different metal bases, we expected significantly different photo-chemical behaviour under the plant’s sunlight radiation, specialised leaf anatomy and over-all plant-response.

Exposure to sunlight can induce ZnO-NPs photo-corrosion, and this enables gradual zinc transport into the plant leaves [2]. Therefore, subsequent ZnO-NPs physiological impact is distinctively different to that of ionic Zn^2+^ and micro-sized zinc species [4]. While TiO_2_-NPs have relatively good photo-stability [7,45] and high mechanical and chemical resistance, O^•^ and OH^•^ radical generation is enhanced when plants are exposed to sunlight, and this can affect plant surface chemistry [45,46]. Although better anatase TiO_2_-NPs light-utilisation and increased associated photosynthesis was reported in spinach in the 400–800 nm visible and ultraviolet spectra [44], other researchers observed cuticle and cell wall damage accompanying higher NPs exposure [22].

Leaf surface analysis showed the presence of non-glandular trichomes (NGTs) and linear glandular trichomes (LGTs) in all variants (Figure 4), but capitate glandular trichomes (CGT) were found only in the ZnO-NPs-treated variant (Table 4).

In addition to the differences in trichomes diversity for each treatment, there was also significant individual leaf trichomes variation in the treatments (Table 4) and a distinct characteristic trend in NGT width and LGT length for the variants in the following order: TiO_2_-NPs > control > ZnO-NPs (Table 5 and Table 6).

Li et al. [2] recorded a significant NGT function in sunflower foliar zinc absorption. Although this is the first report of such trichome association with TiO_2_-NPs, the wider NGTs and longer LGTs than those in leaves collected from ZnO-NPs treated plants indicate the specific sunflower response to TiO_2_-NPs exposure.

### 3.2. Effects of Titanium Dioxide and Zinc Oxide Nanoparticles on the Sunflower’s Quantitative and Nutritional Parameters

Metal oxides such as ZnO, TiO_2_, CuO, and Al_2_O_3_ are used in nano-fertilisers to boost crop growth [36], and ZnO nano-fertilisers particularly provide an alternative to conventional chemical fertilisers by introducing the micro-nutrients required for efficient plant growth and development [71,72,73]. General zinc uses include its benefits in catalytic activity; such as its function in dehydrogenases, aldolases, isomerases, transphosphorylases and RNA and DNA polymerases. Zinc is also important in tryptophan synthesis, cell division, and maintaining membrane structure and potential, and it is beneficial in both photosynthesis and as a regulatory cofactor in protein synthesis [74]. Moreover, zinc has previously been recorded herein as an essential micronutrient for plant growth and development [75], and our research into its application to the Neostar hybrid sunflower cultivar recorded greater resistance to both drought and water-logging. Zinc therefore has a most important function in our hybrid’s production process because the ZnO nanoparticles are quickly transported into the sunflower and participate in its metabolic processes [76].

TiO_2_, SiO_2_, and carbon nanotubes are also part of the new generation of nanoparticle fertilisers. While these have promoted plant growth, controversy surrounds their use because of their potential toxicity. However, beneficial results have been reported by Lu et al. [77] who recorded increased nitrogen fixation in *Glycine maximum* and improved seed germination and growth using a TiO_2_ and SiO_2_ mixture and Gao et al. [78] demonstrated that TiO_2_ alone increased total nitrogen, protein, and chlorophyll content in the *Spinacia oleracea* species.

Our results revealed no statistically significant difference in the number of plants and heads between the plants sprayed with ZnO-NPs and the NP-free controls. This was expected because no significant negative effects of visible leaf damage, growth inhibition, or decreased yield have ever been detected with NPs foliar application [79]. The TiO_2_-NPs also had no negative effect on the number of plants or seed heads, and harmful effects with foliar TiO_2_-NP application only occur at concentrations over 1000 mg·kg^−1^ [19].

Khater [41] reported that 6 mg·L^−1^ TiO_2_-NPs spray-application to *Coriandrum sativum* L. increased the plant height, number of branches and fruit yield, and Moaveni et al. [48] added the positive effect on *Hordem vulgare* L. yield. These results support our statistically significantly increase in head diameter, dry-seed head weight, grain yield, and TSW. The positive effect of 0.02% TiO_2_-NPs on *Triticum aestivum* L., cv. ‘Pishtaz’ spring wheat TSW weight was also confirmed by Jaberzadeh et al. [3], in their highlights of TiO_2_ benefits. Although our previous research in Kolenčík et al. [39] indicated little to no ZnO-NPs effect on foxtail millet quantitative parameters, the positive effects of ZnO-NPs treatment in this research was statistically significant compared to the control. While this was noted in head diameter, dry-seed head weight, yield and TSW, there was less growth improvement than in the TiO_2_-NPs-treated variant. Hussain et al. [33] also reported that ZnO-NPs increased yield, wheat growth, and dry weights after NPs foliar application under Cd stress condition. The ZnO-NPs there most likely increased plant zinc distribution, with consequent Cd stress reduction. Additional authors have also confirmed ZnO-NPs beneficial impact on plant growth parameters [11,24,50].

Statistically significant differences were also found in nutritional parameters. This was especially apparent in terms of the increased oil content, which was 63.6% in the TiO_2_-NPs variant compared to 59.2% in the control. Although the 60.5% sunflower oil content in the ZnO-NPs-treated variant was only slightly higher than the control, this increase was valuable (Table 7). These results are important for agriculture science and quite impressive when compared to 44% USDA seed oil content [80], and the recent FAO report of up to 50% in new Russian sunflower variety seeds [81].

The superior TiO_2_-NPs variant performance was unexpected because these are usually considered only plant-growth enhancers and do not have the overall ZnO-NPs nanofertiliser effects [15]. Raliya et al. [19] also implied this phenomenon in their report of higher tomato lycopene production by TiO_2_-NPs use compared to ZnO-NPs. Herein, we established that foliar application of both ZnO-NPs and TiO_2_-NPs have positive effects on the common sunflower physiological parameters, plant growth and seed quality. Especially in the case of ZnO-NPs, the full assimilation by plant and transformation to chelated Zn species, including zinc-citrate, -oxalate and -phytate [2] and their subsequent transport into seeds would be beneficial for food industry.

In contrast, Figure 5 shows no significant difference in zinc and titanium concentrations in treated fully-ripe seeds compared to the control. However, we consider that this is most likely due to the low-level NPs concentrations applied.

### 3.3. Sunflower Physiological Response to Titanium Dioxide and Zinc Oxide Nanoparticle Foliar Application

Authors recorded that nano-fertilisers positively affect plant growth, flowering and fruit maturation [19,32] and they also provide protection in negative seasonal conditions [82,83], such as salt stress and drought [24,84]. Figure 6 shows that local weather dynamics in the 2018 growing season indicated greater than normal drought.

However, chlorophyll content increases when the sunflower is subject to the foliar application of suitable concentrations of crystalline anastase TiO_2_-NPs in adequate environmental conditions. These conditions include sunlight, higher temperature and appropriate precipitation, and the enhanced chlorophyll content then enables plants to synthesise more light-harvesting pigment-protein complexes (LHCII) which absorb greater light energy. Modification was employed because nano-size anatase encourages oxidation-reduction under specific light wavelengths, and this enhances charge transfer between the nanoparticles and LHCII with resultant photosynthesis increase [85]. In support of this, Lei et al. [44] identified that TiO_2_-NPs under visible and ultra-violet radiation significantly enhances the entire chlorophyll chain electron transport, photosystem II reduction, oxygen production and photo-phosphorylation.

The average normalized difference vegetation index value (NDVI) generally reflects leaf chlorophyll content, photosynthesis-activity, stomatal volume and transpiration [86,87]. While Table 7 shows that this had the following trend: ZnO-NPs variant > TiO_2_-NPs variant > control, Figure 7a highlights that it was statistically significant only for ZnO-NPs at plant ripening on the 125th experimental day. This indicates a vital and prolonged vegetation period from ZnO-NPs application, and early plant maturation with lower chlorophyll content for plants exposed to TiO_2_-NPs. There were also statistically significant differences at stem elongation and flower-bud formation on the 70th day for both TiO_2_-NPs and ZnO-NPs compared to the control, and statistically significant difference between the TiO_2_-NPs variant and control at seed development on the 105th day (Figure 7a).

The photo-chemical reflectance index (PRI) indicates increased photo-chemical activity and photosynthesis [87]. It also shows the assimilation of photosynthetic light-use-efficiency normally linked to the de-exposition stage of xanthophyll cycle pigments which protect the photosynthetic apparatus against photo-damage [88].

Table 7 shows that this is followed by analogous NDVI trend with total seasonal-measured values, and Figure 7b demonstrates the effect at the ripening phase. The results, however, indicate it was only statistically different for the ZnO-NPs variant compared to the control. Figure 7b also highlights the high assimilation efficiency of the ZnO-NPs variant compared to both TiO_2_-NPs and control. However, the apparent decrease in PRI value was also observed at seed development where both ZnO-NPs and TiO_2_-NPs-treated plants had statistically significantly higher values than the control.

Finally, the total average of all crop water stress indices (CWSI) measured throughout the whole season demonstrate better values for the ZnO-NPs variant than for both TiO_2_-NPs and control (Table 7). Figure 7c then illustrates that periodical CWSI ZnO-NPs variant values at ripening were statistically significantly higher than for the control. While this result for ZnO-NPs was unexpected, the TiO_2_-NPs variant values most likely reflect early plant maturation.

There was statistically significant improvement in the common sunflower physiological parameters at stem elongation and flower-bud formation stage two weeks after the first foliar TiO_2_-NPs application. Similarly, the second application of TiO_2_-NPs led to plant positive response at seed development. Hong et al. [43] also reported positive effects, including enhanced light absorption, light energy transformation and protoplast protection which counteracted spinach aging. In contrast, TiO_2_-NPs statistically significantly negatively affected sunflower physiology at the ripening stage. This was most likely related to leaf-yellowing and decreased chlorophyll content and photosynthetic activity. This was not observed in ZnO-NPs treated plants, possibly because of sufficient sunlight radiation or higher air temperature during this stage of the sunflower life cycle (Figure 6).

The TiO_2_-NPs effects are therefore most likely caused by their physical-chemical nature and depend on its photo-corrosion stability and physical-mechanical resistance compared to ZnO-NPs [37,44,45,46]. Finally, foliar application of both TiO_2_-NPs and ZnO-NPs is reported to have beneficial impact on early stem development, flowering and fruit appearance compared to the control; but no earlier fruit maturation was noted [19].

The statistically significant differences between ZnO-NPs treatment and the control emerged one-month after the first foliar application and was apparent at stem elongation and flower-bud formation. However, it was surprising that although the NDVI and PRI values at the end of the experiment indicated positive ZnO-NPs plant effects, water stress significantly increased (CWSI). It appears that the ZnO-NPs-treated variant resulted in prolonged sunflower vegetative phase compared to the TiO_2_-NPs and control variants. The slightly delayed sunflower physiological response to ZnO-NPs may have been caused by the low-level NPs concentration, their rapid uptake and assimilation [2], or low precipitation and other weather conditions during the vegetation season (Figure 6).

Soil condition is an important environmental factor that affects plant physiological parameters. The availability of soil zinc for the plant is mainly influenced by lower, or higher pH and higher clay content and carbonates or natural organic matter [89]. However, our Table 1 highlights no soil zinc deficit at 0.9 mg·kg^−1^, neutral pH (pH = 7), low 1.28% carbon content and approximately 2% humus content. This reflects the Dolná Malanta silt loam haplic Luvisol [53], and the total soil zinc content corresponds to the global average [90]. Moreover, foliar application of ZnO-NPs is most easily assimilated by the plant and freely available for the physiological processes required for prolonged plant vegetation. Our results therefore preclude the sunflower sensitivity to zinc deficiency reported in [91]. Finally, Lyu et al. [40] consider that low titanium soil concentration is beneficial for plant production because it simulates certain enzyme enhancement of essential nutrient uptake like Fe, increases crop yield, and promotes a decrease in stress tolerance.

To the best of our knowledge, there is no literature on sunflower interaction with ZnO-NPs related to a wide variety of physiological parameters and its impact on oil content. The available literature contains only the positive ZnO-NP effects on the quantitative and physiological parameters of winter wheat and maize with direct impact on chlorophyll content and photosynthesis [50,92,93]. This is normally an obligatory requirement to increase oil content [94].

In addition, Kirnak et al. [95] reported that decreased CWSI resulted in higher oil content and Candogan et al. [96] supported this with lower CWSI increasing soy seed-proteins and oil content. Finally, Ahmad et al. [97] found statistically significant differences in *Mentha piperita* essential oil content and quality after their 150 mg·L^−1^ TiO_2_-NPs foliar application, and 90 mg·L^−1^ TiO_2_-NPs concentration improved *Vetiveria zizanioides* photosynthesis and essential oil yield [98].

## 4. Conclusions

Our hypothesis is based on the sunflower’s extensive ability as a photo-sensitive plant to combat sunlight radiation, and here the foliar application of two types of photo-active nano-fertilisers accelerated both predictable and unpredictable plant functions and reactions.

We discovered that the TiO_2_-NPs treated variant improved sunflower quantitative and nutritional parameters including oil content. The TiO_2_-NPs provided unexpected early plant maturation with all dependent physiological indices. These results were most surprising, because although TiO_2_-NP plant enhancers are more photo-stable, they provide less nutrition and are potentially toxic. In contrast, the photo-corrosive and less resistant ZnO-NPs treated variant predicted, and better reflected, the sunflower physiological parameters with relatively good quantitative and nutritional effects. The most likely reason for its superiority is the increased zinc bioavailability following ZnO-NP transformation, and this supports both the sunflower physiology and its metabolic pathways.

There is no doubt that sunflower foliar application of low nanoparticle concentrations far surpasses NP-free control’s agronomic, environmental, and economic benefits. This is clearly evident in the comparison of most plant parameters. There are also significant differences in trichome distribution between the NPs-treatment methods and the control, and this is especially noted in trichome width and length on the surface of leaves collected at the flower-bud development stage. Moreover, neither nanoparticle treatment detrimentally affected final food quality. This was proven by analysis which showed no higher Zn or Ti translocation to fully ripe sunflower seeds compared to the NPs-free control.

In conclusion, although nano-fertilisers are widely used to enhance plant production, yield and fruit quality, their effects remain unpredictable. There are also unexplored issues. These include the NPs fertiliser mode of action and uptake mechanisms and dose-dependent plant response. Agricultural nanotechnology therefore remains one of the most challenging disciplines in combatting current climate change.

## Figures and Tables

**Figure 1 nanomaterials-10-01619-f001:**
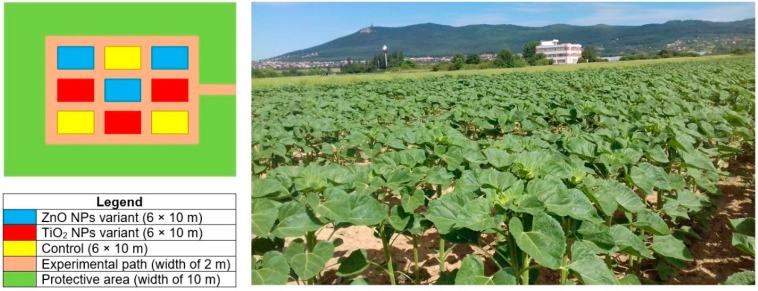
Schematic experimental field plant in aerial view.

**Figure 2 nanomaterials-10-01619-f002:**
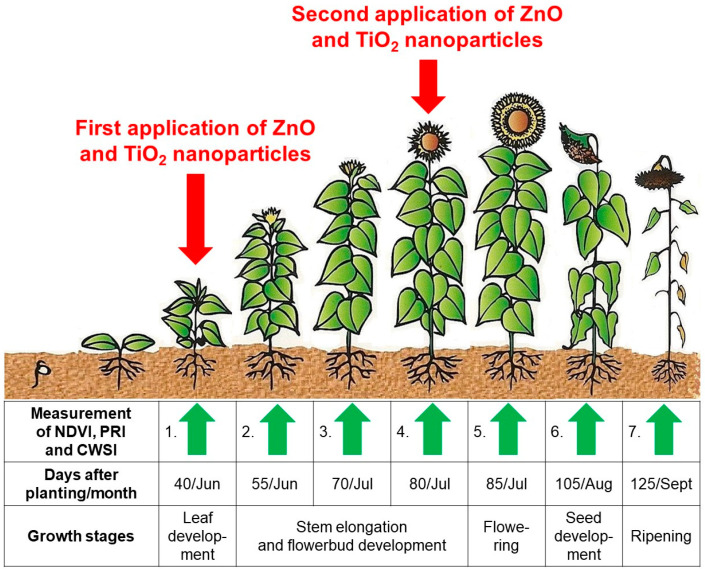
Schematic model of sunflower growth stages. While the red arrows indicate the foliar application of TiO_2_-NPs or ZnO-NPs, the green arrows indicate time intervals for measuring physiological parameters; normalised difference vegetation index (NDVI), photochemical reflectance index (PRI) and crop water stress index (CWSI) for the assessment of sunflower development.

**Figure 3 nanomaterials-10-01619-f003:**
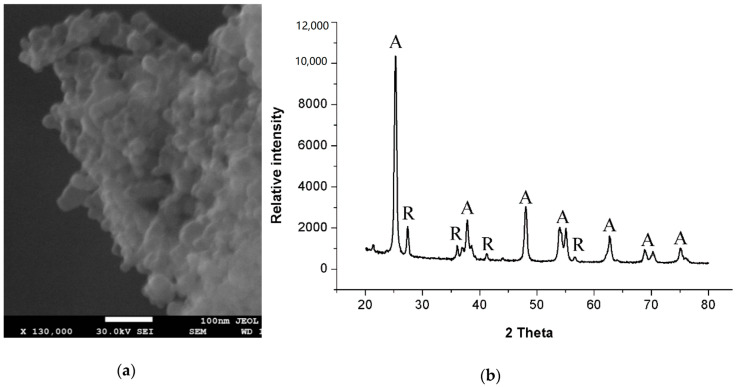
(**a**) Scanning electron microscopy of TiO_2_-NPs, (**b**) X-ray diffraction analysis confirmed that TiO_2_-NPs have anatase (A) and rutile (R) polymorphic modification.

**Figure 4 nanomaterials-10-01619-f004:**
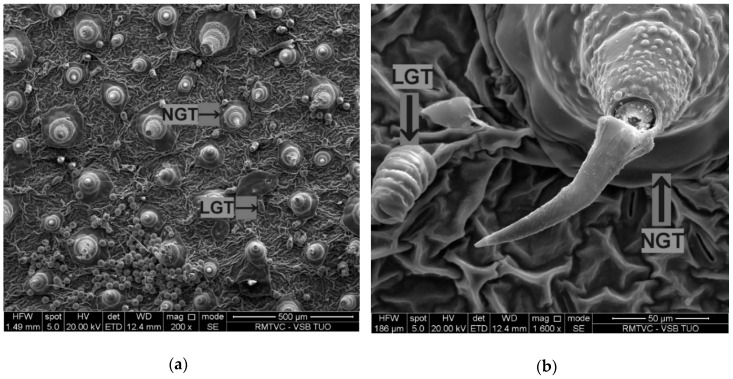
Representative scanning electron image of sunflower leaf surface with non-glandular trichomes (NGTs) and linear glandular trichomes (LGTs) in both NPs-treated variants and the control—(**a**) leaf surface, (**b**) trichome detail.

**Figure 5 nanomaterials-10-01619-f005:**
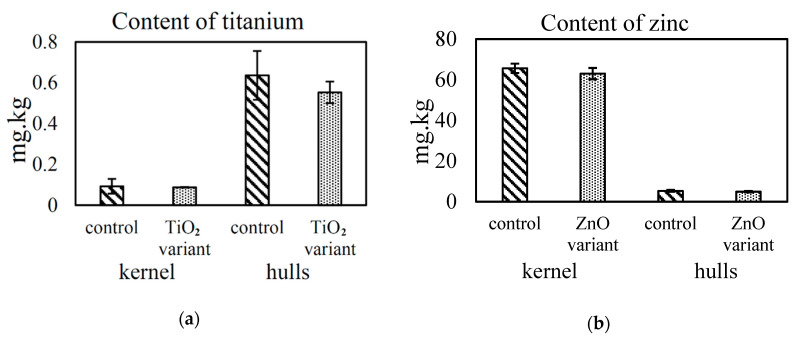
Comparison of total titanium (**a**) and zinc (**b**) concentrations in the kernel and hulls of fully-ripe sunflower seeds after foliar application of titanium dioxide, zinc oxide nanoparticle variants and the control.

**Figure 6 nanomaterials-10-01619-f006:**
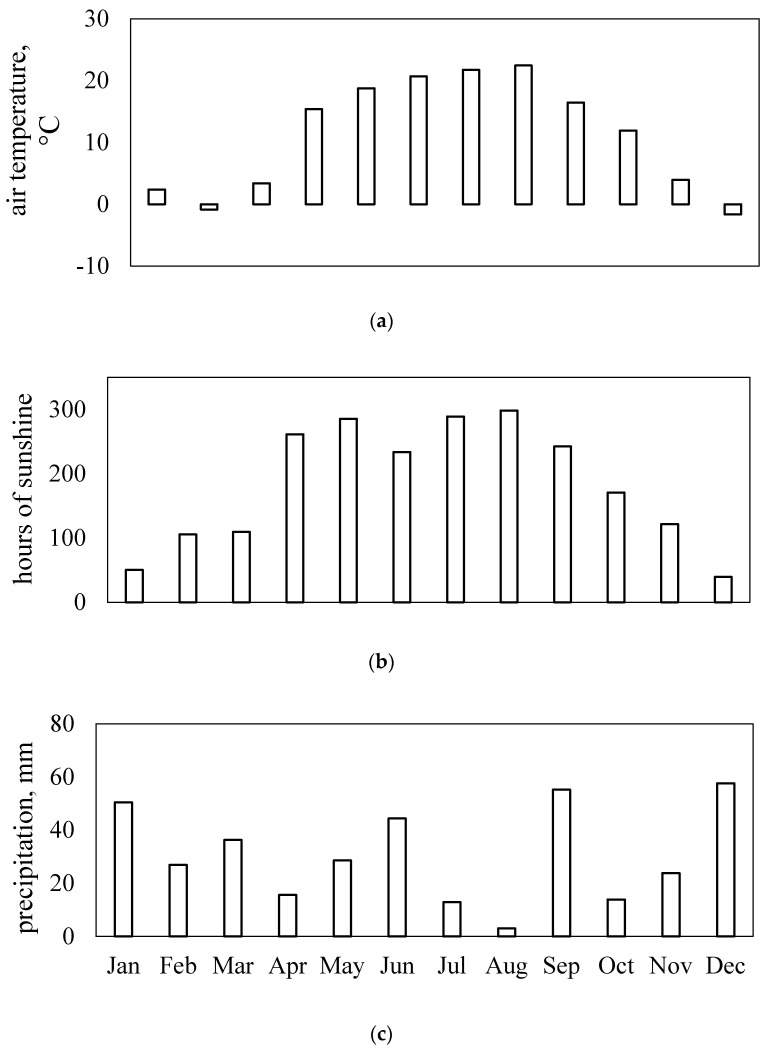
Monthly seasonal variations in the 2018 vegetation season at the Dolná Malanta experimental research field in Nitra in the Slovak Republic; (**a**) air temperature, (**b**) sunshine hours and (**c**) precipitation.

**Figure 7 nanomaterials-10-01619-f007:**
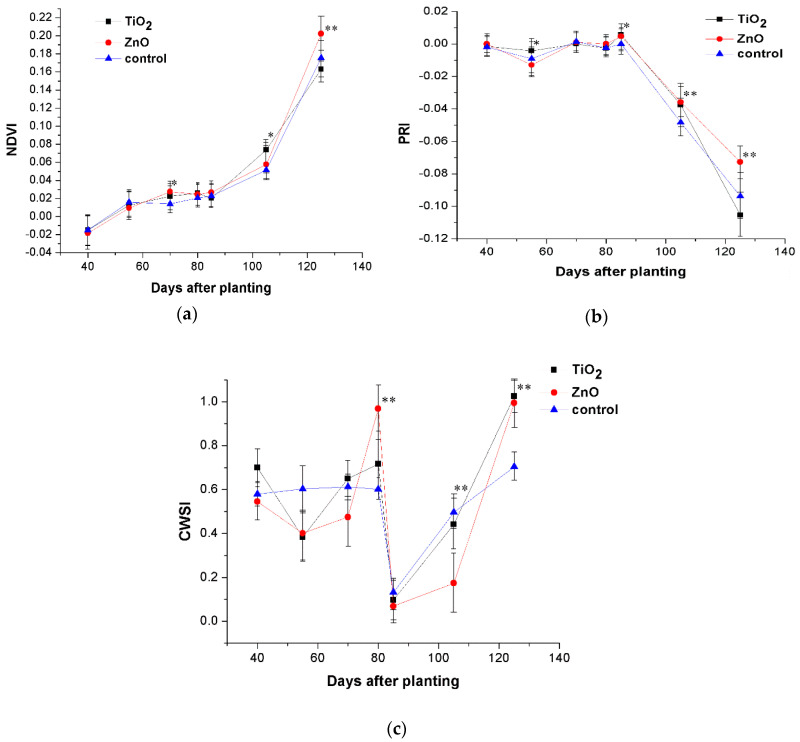
Analysed 2018 seasonal effects of sunflower foliar-sprayed titanium dioxide and zinc oxide nanoparticles variants associated with the following physiological indices and compared to the control; (**a**) normalised difference vegetation index (NDVI), (**b**) photochemical reflectance index (PRI) and (**c**) crop water stress index (CWSI), the significance: * *P* value < 0.05, ** *P* value < 0.01.

**Table 1 nanomaterials-10-01619-t001:** Typical soil characteristic before autumn 2017 sunflower planting in the experimental locality (Dolná Malanta, Nitra, Slovak republic).

Soil Depth (m)	pH	Nutrients Content (mg·kg^−1^) (Mehl.III)								Humus (%)	Carbonates
		N_in_	P	K	Ca	Mg	Na	Zn	Mn		
0–0.3	7.03	7.9	28.8	285	5150	824.5	300	0.90	7.97	2.03	1.28
0.3–0.6	6.93	7.3	18.8	225	2400	801.7	300	0.82	5.23	1.64	0.4

**Table 2 nanomaterials-10-01619-t002:** The nitrogen-related soil characteristics before spring 2018 planting of sunflower in the experimental locality (Dolná Malanta, Nitra, Slovak Republic).

Soil Depth	Nutrient Content (mg kg^−1^)		
	N-NH_4_^+^	N-NO_3_^−^	N_an_
0–0.3	9.5	9.2	18.7
0.3–0.6	10.0	8.4	18.4

**Table 3 nanomaterials-10-01619-t003:** X-ray diffraction analysis of TiO_2_-NPs polymorphic modifications, with estimated symmetry and unit cell parameters.

	Anatase	Rutile
Crystal symmetry	Tetragonal	Tetragonal
*a*-axes	3.7874 ± 0.0001 Å	4.5973 ± 0.0005 Å
*c*-axes	9.5111 ± 0.0007 Å	2.9582 ± 0.0005 Å
α, β, γ	90°	90°
Space Groups	*I*4_1_/*amd*	*P*4_2_/*mnm*
Unit cell volume *	136.2 Å^3^	62.36 Å^3^
L_vol_-IB (nanometers) **	19.6 ± 0.2	30.0 ± 2.0
Content of TiO_2_ NPs polymorphs (%) **	84.25%	15.75%

* calculated from Unit Cell, ** Calculated from X-ray diffraction analysis.

**Table 4 nanomaterials-10-01619-t004:** Distribution of the three trichomes on leaves collected during flower-bud formation after two TiO_2_-NP and ZnO-NPs foliar applications compared to NPs-free controls (N—number of analysed trichomes, NGTs—non-glandular trichomes, LGTs—linear glandular trichomes, CGT—capitate glandular trichomes).

	Control		ZnO-NPs Variant		TiO_2_-NPs Variant	
	N ^1^	%	N ^1^	%	N ^1^	%
NGTs	106	20.9	112	34.3	110	34.3
LGTs	402	79.1	196	59.9	211	65.7
CGTs	-	-	19	5.8	-	-
Summary	508	100	327	100	321	100

^1^ N—number.

**Table 5 nanomaterials-10-01619-t005:** Tukey HSD average μm widths at α = 0.01 for non-glandular trichomes on leaves collected during flower-bud formation after two foliar TiO_2_-NPs and ZnO-NPs applications and the NP-free control (Min—minimum, Max—maximum, C_v_—coefficient of variation).

	Control	ZnO-NPs	TiO_2_-NPs
Average (μm)	76.5 ± 14.7	57.6 ± 15.6	80.1 ± 22.8
Min	52.7	35.2	46.9
Max	120.2	111.5	161.2
Number	100	100	100
C_v_	19.2	27.2	28.4

**Table 6 nanomaterials-10-01619-t006:** Tukey’s HSD average μm lengths at α = 0.01 for linear glandular trichomes on leaves collected during flower-bud formation after two foliar TiO_2_-NP and ZnO-NP applications, and the NP-free control (Min—minimum, Max—maximum, C_v_—coefficient of variation).

	Control	ZnO NPs	TiO_2_ NPs
Average (μm)	52.9 ± 5.0	49.3 ± 6.4	56.4 ± 5.7
Min	41.4	38.2	43.3
Max	65.5	61.0	70.6
Number	100	50	50
C_v_	9.4	12.9	10.1

**Table 7 nanomaterials-10-01619-t007:** List of TiO_2_-NPs- and ZnO-NPs-treated sunflower plant quantitative, nutritional and physiological parameters, and those for the control. Plants were harvested after 125 days. Results show standard deviation and Fisher’s least significant difference.

	TiO_2_-NPs Foliar Applied Variant	ZnO-NPs Foliar Applied Variant	Control Variant (No NPs Application)
*Quantitative parameters*			
Number of plants per hectare	77442 ± 237 ^ns^	77548 ± 366 ^ns^	77658 ± 66
Number of head per hectare	77851 ± 341 ^ns^	77687 ± 423 ^ns^	78007 ± 117
Head diameter (mm)	314 ± 9 **	276 ± 9 **	265 ± 9 *
Weigh of dry seed head (g)	231.01 ± 13.73 **	191.1 ± 20.9 **	160.2 ± 19.0
Weight of thousand seeds (TSW) (g)	76.11 ± 7.38 *	66.6 ± 1.7 *	53.2 ± 10.9
Grain yield (t·ha^−1^)	4.50 ± 0.28 **	3.4 ± 0.5 **	2.5 ± 0.3
*Nutritional parameter*			
Content of oil (%)	63.64 ± 1.59 *	60.5 ± 1.1 *	59.2 ± 1.0
*Physiological parameters*			
NDVI ^1^	0.043 ± 0.004 ^ns^	0.047 ± 0.005 ^ns^	0.041 ± 0.005
PRI ^2^	−0.021 ± 0.002 **	−0.017 ± 0.001 **	−0.022 ± 0.003
CWSI ^3^	0.57 ± 0.04 ^ns^	0.52 ± 0.10 ^ns^	0.53 ± 0.06

^1^ Normalised difference vegetation index (NDVI), ^2^ Photochemical reflectance index (PRI), ^3^ Crop water stress index (CWSI), the significance: * *P* value < 0.05, ** *P* value < 0.01, ^ns^ Non-significant.

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
