# Peer review of "Foliar Application of Low Concentrations of Titanium Dioxide and Zinc Oxide Nanoparticles to the Common Sunflower under Field Conditions"

_nanomaterials, 2020, doi:10.3390/nano10081619_

Round 1

Reviewer 1 Report

Τhis is a thorough work on the effect of nanofertilizers on sunflower oil production.

The issue is that beyond the experiments and the conclusions drown from them, there is still no approach to the issue of nanoparticle interaction with plant tissue and overall mechanism of activity.

I suggest inclusion of such experiments and results to the work that would greatly influence its scientific impact.

Also please check English language for minor mistakes such as

line 33 Foliarly applied nanoparticles were of well crystallinity with ..

line 105 ..the officially research field of Slovak University of Agriculture..

Author Response

Review 1.

  1. The issue is that beyond the experiments and the conclusions drown from them, there is still no approach to the issue of nanoparticle interaction with plant tissue and overall mechanism of activity.

# Answer:

  • Presented manuscript was not primarily intended to evaluated the mechanisms of interaction between NPs with plant tissue and overall mechanism of activity, although there were involved/ added several associations related to NPs-leaf surface activity, and responds, or basic relationship to mechanisms, e.g. L.230 “…other research observed cuticle and cell wall damage accompanying higher NP exposure.. [61]., or L. 350“This was most likely related to leaf-yellowing and decreased chlorophyll content...”, L. 66-68 “Zinc is a cofactor for over three hundred enzymes and has important functions in gene expression and protein synthesis, and it also structurally and functionally stabilises membranes and proteins [4, 16, 30]”, “Although better anatase TiO2-NP light-utilisation and increased associated photosynthesis was reported in spinach in both the 400-800nm visible and ultraviolet spectra”.
  • Additionally, hypothesis of manuscript was expanded about micronutrient-related fertilizer ZnO-NPs in comparison with plant-growth enhancer TiO2-NPs, and discussed at different parts of manuscript L.53-58. “Nanoparticles are also extremely useful in agriculture where they can alleviate the effects of plant diseases and are active components in nano-fertilisers [11-18]. Liu and Lal [15] classify nano-fertilisers in the following categories: (i) macro-nutrient nano-fertilisers; (ii) micro-nutrient nano-fertilisers that include nanoparticles which are mainly oxides such as ZnO, CuO, Fe2O3, etc.; (iii) nutrient-augmented nanomaterials, such as zeolites, and (iv) plant-growth enhancers with unpredictable action, such as TiO2 and carbon nanotubes.”
  1. I suggest inclusion of such experiments and results to the work that would greatly influence its scientific impact.

# Answer:

  • Regarding the fact that field experiment was conducted more than 5-6 months it is no possible repeated it with analogical conditions.
  1. Also please check English language for minor mistakes such as

line 33 Foliarly applied nanoparticles were of well crystallinity with ..

line 105 ..the officially research field of Slovak University of Agriculture..

# Answer:

  • English language correction of manuscript was controlled by English lector Dr. Ray. J. Marshall.

Reviewer 2 Report

This manuscript investigates the effects of folliarly applied nanoparticles on sunflower plant performance, finding that this way of fertilization does not negatively affect seed composition and it results in benefits to the plant.

The study aims at f evaluate the effects of foliar application of both ZnO-NPs and TiO2-NPs over one season on common sunflower. The study includes various qualitative and quantitative parameters of plants and seeds, and morphological properties of leaves and metal contents in fruits. This is a novel and interesting study, that brings insights into alternative ways of crop fertilization

The abstract should include a sentence justifying the use of Zinc and Titanium. Similarly, the results included in the abstract should be more precise and not so vaguely presented.

The introduction, although short, is clear and informative. However, the actual objectives of the study are obscurely presented and there is not scientific hypothesis to be tested. As this is an experimental study, hypothesis is a must. Please, include them.

It is unclear to me why was it needed to Characterization of titanium dioxide and zinc oxide nanoparticles. ¨They were purchased and hence one would have assumed that their properties would have been provided by the manufacturer.

In the material and methods section it is explained the sunflower variety used, the amount of fertilizers (NPK) added to the soil, as well as the use of a fungicide. It is necessary that the authors explain why were those nutrients added, why was that the variety of sunflower used and the same for the fungicide and how it can interfere with the nanoparticle application. Similarly, it is important to explain why there were two applications of nanoparticle fertilization and why it was apply at the two different phonological stages.

On line 176 it is said ‘The translocated and accumulated zinc and titanium in the sunflower seeds’. However, nothing is explained how translocation was estimated.

What was the procedure to calculate photochemical reflectance index, normalized difference vegetation index, and crop water stress index physiological parameters? It is included the name of the deviced that measures photochemical reflectance, but nothing is said on how may leaves were measured, for how long and at what time of the day the data were collected. The same can be said for the crop water stress index.

Did you test if the data met the needs for an ANOVA? If so, how?

Trichomes were measured only in treated plants, so we do not know if the shape of them is a consequence of the nanoparticle application or if all trichomes are similar in any leaf of sunflowers of the same variety.

Line 257, change to ‘no significant statistical differences were detected in….’

The discussion is mostly descriptive and the authors compare their results with results from others. However, there is no explanation on why this species benefits from the application of the nanoparticles. There is close to no integration of all the results and how they together explain the behaviour of the plants. This end has to be fixed and the authors have to build a more elaborated discussion that goes beyond mere descriptions. How do the soil nutritional parameters affect plant performance? What are the putative implication of the environmental variables?

The presented conclusions are not such. This section, in its current form is a mere summary of the results, with an extended attempt to develop the poor discussion. Should you have scientific hypothesis properly depicted, the conclusions would flow nicely. Please, rewrite the entire conclusions section to make it a proper conclusive paragraph instead of a reiteration of the results, as indicated.

Author Response

Review 2.

  1. The abstract should include a sentence justifying the use of Zinc and Titanium. Similarly, the results included in the abstract should be more precise and not so vaguely presented.

# Answer: Agree, several points were changed/ added include L. 32-35 “The benefits arising from foliar application of micronutrient-based zinc oxide fertiliser were compared with those from the titanium dioxide plant-growth enhancer. Both the zinc oxide (ZnO) and titanium dioxide (TiO2.) were delivered by foliar application in nano-size at concentration of 2.6 mg.L-1.” or, L.40-41 “…and changed sunflower physiology to early maturation...”

  1. The introduction, although short, is clear and informative. However, the actual objectives of the study are obscurely presented and there is not scientific hypothesis to be tested. As this is an experimental study, hypothesis is a must. Please, include them.

# Answer: Agree, the hypothesis of manuscript was expanded about micronutrient-related fertilizer ZnO-NPs in comparison with plant-growth enhancer TiO2-NPs, and discussed at different parts of manuscript L.53-58. “Nanoparticles are also extremely useful in agriculture where they can alleviate the effects of plant diseases and are active components in nano-fertilisers [11-18]. Liu and Lal [15] classify nano-fertilisers in the following categories: (i) macro-nutrient nano-fertilisers; (ii) micro-nutrient nano-fertilisers that include nanoparticles which are mainly oxides such as ZnO, CuO, Fe2O3, etc.; (iii) nutrient-augmented nanomaterials, such as zeolites, and (iv) plant-growth enhancers with unpredictable action, such as TiO2 and carbon nanotubes.” L.87-89 “Our hypothesis therefore compares the effects of foliar application of micro-based ZnO-NPs fertiliser and combine two-polymorphs of TiO2-NPs plant-growth enhancer on the Helianthus annuus L. common sunflower over one season.

  1. It is unclear to me why was it needed to Characterization of titanium dioxide and zinc oxide nanoparticles. ¨They were purchased and hence one would have assumed that their properties would have been provided by the manufacturer.

# Answer:

  • According to our long-term empirical investigation with similar commercial products, we know that declaration it usually not enough, they “only” present “size approximation e.g under 50 nm, or chemical stoichiometry composition e.g. TiO2-NPs” without particular species (polymorphic modification with specific crystallographic determination) which indeed could have different plant-effect. Due to these, we present in manuscript relevant information, e.g. L. 211-215 “This morphology corresponds to the rutile and anatase TiO2 polymorphic modifications identified by XRD analysis (Figs. 3a and 3b). The mean size of the anatase and rutile crystals is approximately 19.6±0.2 and 30.0±2.0 nm, respectively (Tab. 3). The X-ray diffraction analysis in Table 3 also revealed both rutile and anatase have tetragonal symmetry and different unit cell parameters and relative content.”, or L. 228-230 “Although better anatase TiO2-NP light-utilisation and increased associated photosynthesis was reported in spinach in both the 400-800nm visible and ultraviolet spectra [39], other research observed cuticle and cell wall damage accompanying higher NP exposure [61]”. L. 86-88 “Our hypothesis therefore compares the effects of foliar application of micro-based ZnO-NPs fertiliser and combine two-polymorphs of TiO2-NPs plant-growth enhancer on the Helianthus annuus L. common sunflower over one season.”
  1. In the material and methods section it is explained the sunflower variety used, the amount of fertilizers (NPK) added to the soil, as well as the use of a fungicide. It is necessary that the authors explain why were those nutrients added, why was that the variety of sunflower used and the same for the fungicide and how it can interfere with the nanoparticle application. Similarly, it is important to explain why there were two applications of nanoparticle fertilization and why it was apply at the two different phonological stages.

# Answer:

The aim of the manuscript was simulated the standard procedures of cultivation of sunflower in real-field condition. Due to this fact, we applied widespread European most common sunflower with all “traditional” agrotechnical processes (application of herbicides, fungicides, fertilization, etc.), where usually two times foliar application of every fertilization (including nanoparticles) take place.

All herbicides, and fungicides belong to standardized chemicals route applied at minimal concentration without long-term environmental effect as described in manuscript L.139-148. ”Combined-type fertilisers containing 15% nitrogen, 15% P2O5 and 15% K2O (ACHP Levice, a. s., Levice, Slovak Republic) were applied by tractor as a pre-sowing soil treatment using the Ferti fertiliser applicator (Agromehanica, Boljevac, Serbia). The fertilisers were applied in 200 kg.ha-1 concentration based on soil agrochemical analysis before planting. The experimental SY Neostar sunflower hybrid was sown in lines with 60mm sowing depth, 220mm seed distance and 700mm inter-row spacing by the Monosem NG Plus 3 planter (Monosem, Largeasse, France) [55]. 4 L.ha-1 of Wing® P herbicide (BASF, Ludwigshafen am Rhein, Germany) was applied pre-emergently and 0.5 L.ha-1 Pictor® fungicide (BASF, Ludwigshafen am Rhein, Germany) was used 55 days after planting. All replications, including controls, were treated with herbicide and fungicide by AGT 865T/S sprayer (Agromehanica, Boljevac, Serbia).” Generally, application of leaf fertilizers, or leaf-stimulators usually take place two times for vegetation season (around 40th days, and 80th days) as mentioned in manuscript L.149-154„Plants were sprayed with 2.6 mg.L-1 TiO2-NPs or 2.6 mg.L-1 ZnO-NPs dispersed by GAMMA 10 hand sprayer (Mythos Di Martino, Mussolente, Italy). This was performed on early and windless mornings, and until the leaves were completely wet. Foliar application was conducted on the 40th day when the sunflower reached the phenological growth phase of leaf development and again on the 80th day when there was stem elongation with flower-bud formation; as in Meier [56] (Fig. 2). The NP-free controls had only water spraying.“

It is a matter of fact that herbicide and fungicide, and fertilizers were applied at completely different times, and for all of variant we do not expected any interfere to/ with NPs-related effect of application. 

  1. On line 176 it is said ‘The translocated and accumulated zinc and titanium in the sunflower seeds’. However, nothing is explained how translocation was estimated.

# Answer:

This information has been already described in manuscript L.182-185. “…the total zinc and titanium concentrations were determined by ICP-MS (Thermo Scientific iCap-Q, Germany) in KED mode with helium gas; the calibration solutions were prepared from MERCK CertiPUR ICP 1000 mg.L-1 single-element standard solutions (Germany) and scandium and rhodium provided internal standards.

  1. What was the procedure to calculate photochemical reflectance index, normalized difference vegetation index, and crop water stress index physiological parameters? It is included the name of the deviced that measures photochemical reflectance, but nothing is said on how may leaves were measured, for how long and at what time of the day the data were collected. The same can be said for the crop water stress index.

# Answer:

We agree, information with relevant citations were added to manuscript L.194-203 “Non-destructive methods were ensured for all measurements, and these were performed from 11.00 a.m. to 1.00 p.m. on similar dates, plants and growth phases. Each PRI and NDVI repetition on ten different annual sunflower mature leaves was labelled and measured during the growing season. Here, we used at least ten perpendicularly-oriented leaf measurement points for each index in order to cover all the leaf heterogeneity described by Gamon et al. [60].

The crop water stress index (CWSI) was calculated according to Jones, et al. [61]. Measurement of atmospheric moisture, leaf temperature and dry and wet leaf surface were required for the CWSI index calculation, and these were provided by EasIR-4 thermo-camera (Bibus AG, Fehraltorf, Switzerland). The thermal images gave diagonal sunflower scanning from 2m distance, 1.5m height and 20.6° x 15.5° auto-focus field of view.

  1. Did you test if the data met the needs for an ANOVA? If so, how?

We agree, information were specify according to reviewer suggestion, L.205-208 “Prior to evaluation of the multifactorial analysis of variance (ANOVA), the normality of experimental data was tested at α = 0.05 and α = 0.01 significance by the following: the Student t-test, Shapiro-Wilkov test for trials and Fisher’s least difference (LPS)”.

  1. Trichomes were measured only in treated plants, so we do not know if the shape of them is a consequence of the nanoparticle application or if all trichomes are similar in any leaf of sunflowers of the same variety.

# Answer:

This information has been mentioned in manuscript on different place, e.g. Tab. 4, 5, 6. In experiment, we applied one type of sunflower cultivar “Neostar” L. 101-103 “The experimental SY Neostar hybrid of the Helianthus annuus L. common sunflower (Syngenta, Basel, Switzerland) is part of the two-line imidazolinone-resistant hybrid suitable for the ClearField Plus® production system.”, L. 388-390 “Herein, we evaluated the effects of low concentration TiO2-NP and ZnO-NP foliar application on the Helianthus annuus L. common sunflower. This is one of the most widespread “Neostar” cultivars sown in European field conditions.”

  1. Line 257, change to ‘no significant statistical differences were detected in….’

# Answer:

This sentence were re-written (see Tab. 7) L.”There was no statistically significant difference in the number of plants and heads for plants sprayed with ZnO-NPs and NP-free controls.”.

  1. The discussion is mostly descriptive and the authors compare their results with results from others. However, there is no explanation on why this species benefits from the application of the nanoparticles. There is close to no integration of all the results and how they together explain the behaviour of the plants. This end has to be fixed and the authors have to build a more elaborated discussion that goes beyond mere descriptions. How do the soil nutritional parameters affect plant performance? What are the putative implication of the environmental variables?

# Answer:

We agree, the soil nutrional parameters as another environmental factor (variables) were added in manuscript with relevant references, L. 367 – 376 “Soil condition is an important environmental factor that affects plant physiological parameters. The availability of soil zinc for the plant is mainly influenced by lower, or higher pH and higher clay content and carbonates or natural organic matter [71]. However, our Table 1 highlights no soil zinc deficit at 0.9 mg.kg-1, neutral pH (pH=7), low 1.28% carbon content and approximately 2% humus content. This reflects the Dolná Malanta silt loam haplic Luvisol [48], and the total soil zinc content corresponds to the global average [72]. Moreover, foliar application of ZnO-NPs is most easily assimilated by the plant and freely available for the physiological processes required for prolonged plant vegetation. Our results therefore preclude the sunflower sensitivity to zinc deficiency reported in [73].

  1. The presented conclusions are not such. This section, in its current form is a mere summary of the results, with an extended attempt to develop the poor discussion. Should you have scientific hypothesis properly depicted, the conclusions would flow nicely. Please, rewrite the entire conclusions section to make it a proper conclusive paragraph instead of a reiteration of the results, as indicated.

# Answer: We agree, scientific hypothesis was added, and conclusion was completely re-write to more logical framework, L. 386-411 “Herein, we evaluated the effects of low concentration TiO2-NPs and ZnO-NPs foliar application on the Helianthus annuus L. common sunflower. This is one of the most widespread “Neostar” cultivars sown in European field conditions. However, our spraying of the micronutrient ZnO-NPs fertiliser and TiO2-NPs plant-growth enhancer twice during the plant life cycle brought unexpected results.

The physiological parameters of ZnO-NPs treated variant were determined by the following indices: normalised difference vegetation, photochemical reflectance and crop water stress. The expected positive results did not correspond with the determined quantitative and nutritional values, because the TiO2-NPs treated plants performed better than those sprayed with ZnO-NPs. This inconsistency may be due to the prolonged ZnO-NPs variant vegetation period and relative suitable physiological response of TiO2-NPs during whole vegetation season, however, with unexpected earlier plant maturation and ripening.

This was off-set by the surprising and contrasting effect of the TiO2-NPs’quantitative and nutritional parameters being a little higher than those of the ZnO-NPs treated plants, but the foliar application of both NPs affected parameters very positively compared to the NPs-free control.

There were also additional significant differences between both treatment methods and the control in trichomes distribution and their width and length on leaf surfaces collected at the flower-bud development stage. However, NPs foliar application did not affect zinc and titanium transport in the kernel and hulls of the fully ripe sunflower seeds.

In conclusion, although nano-fertilisers are widely used to enhance plant production, yield and fruit quality, their effects remain unpredictable. There are also unexplored issues. These include the NPs fertilizer mode of action and uptake mechanisms and dose-dependent plant response. Agricultural nanotechnology therefore remains one of the most challenging disciplines in combatting current climate change.

Reviewer 3 Report

Dear Editor,

The paper is interesting and evidences some positive effects on plant performance due to the application of titanium dioxide and zinc oxide nanoparticles. Therefore, I suggest the acceptance of this manuscript after a minor revision. In particular, data of tables and figures should be completed inserting standard deviation (or standard error). Besides, I recommend to the authors a revision of the English grammar by a native speaker.

The abstract is well organized.

General issues of the ABSTRACT section: please check the English language.

Line 31: it may be field conditions?

Line 31: please insert the concentrations

Line 32: the growth

Line 37: generally had

Line 39: were instead of was

The introduction is untestable and well-focused on the objective of this research. I would suggest an English-language check.

Line 66: Torabian et al. [27] reported it even helps to ameliorate salt stress.. please indicate the species on which such an effect was observed.

Also the section Results and discussion should be checked for the English language.

Author Response

Review 3.

  1. General issues of the ABSTRACT section: please check the English language.

Line 31: it may be field conditions?

Line 31: please insert the concentrations

Line 32: the growth

Line 37: generally had

Line 39: were instead of was

The introduction is untestable and well-focused on the objective of this research. I would suggest an English-language check.

Also the section Results and discussion should be checked for the English language.

# Answer: All body of manuscript were re-written according to reviewer comments, and suggestions. English language correction of manuscript was controlled by English lector Dr. Ray. J. Marshall.

  1. Line 66: Torabian et al. [27] reported it even helps to ameliorate salt stress.. please indicate the species on which such an effect was observed.

Answer: We agree, information were added to manuscript L. X “and Torabian et al. [29] report it helps ameliorate salt stress in cultivars such as “Olsion”....

Round 2

Reviewer 1 Report

Ιt is totally understandable that the scope of the paper is the use of nanoparticles in sunflower production. As the science part concerning the mechanism of interaction - which exact properties cause the effects, why not micro size counterparts etc - is lacking, the paper would be more appropriate for an agriculture journal. For the current journal I would see a much more thorough focus on the scientific mechanism part, rather than the application part.

Author Response

Reviewer 1

  1. Ιt is totally understandable that the scope of the paper is the use of nanoparticles in sunflower production. As the science part concerning the mechanism of interaction - which exact properties cause the effects, why not micro size counterparts etc - is lacking, the paper would be more appropriate for an agriculture journal. For the current journal I would see a much more thorough focus on the scientific mechanism part, rather than the application part.

Answer:

Aims of work was quantify the foliar application with wide-variety effects of two type of nanoparticles to common sunflower, but, kindly thanks for inspirational suggestion for evaluation of the similar macro-particle counterparts, we will consider it further years. However, our type of research was closely based on nanoparticles-related effects, and if we would like to extend it about corresponding macro, or ionic-related parts we do not know reproduce i) all of environmental conditions and experimental time-period currently, ii) if we did it, manuscript will be completely time-consuming, and iii) for a most part will transform to another kind of research based on “classical” agricultural fertilizer-type work which was not intended of our aims. Additionally, macro-size effect will have not any adding value & novelty only comparable background. The several nano-macro-ionic effects have been already mentioned in manuscript, e.g. L.47-49 “Nanoparticles (NPs) are defined as chemical entities with at least one of their three dimensions less than 100 nm and have significantly different physical, chemical and biological properties to their macro-sized and dissolved ionic counterparts...”, L. 259-261 “Exposure to sunlight can induce ZnO-NPs photo-corrosion, and this enables gradual zinc transport into the plant leaves (Li et al. 2019). Therefore, subsequent ZnO-NPs physiological impact is distinctively different to that of ionic Zn2+ and micro-sized zinc species (Mousavi Kouhi et al. 2014).”, L. 353-355“Especially in case of ZnO-NPs, the full assimilation by plant and transformation to chelated Zn species, including zinc-citrate, -oxalate and -phytate (Li et al. 2019) and their subsequent transport into seeds would be beneficial for food industry.

Regarding aims and scope of our manuscript, we choose “Nanomaterials” journal intently due to enormous absent of this topic there (what we pointed in CL as adding value and novelty). Our previous study (Kolenčík et al. 2019) has been published here, and seem to us have wide-readers citation potential. Also, “Nanomaterials” (according to authors instruction) do conventual non-limited our type of results. Research under field conditions are often many times harder, and time-dependent than NPs + strictly lab-regulated or green-house studies of seed, or early plants development which seem to us is currently relative enough published in literature, and this fact is also pointed with fruit quality issue in manuscript L.88-91. “Most referenced papers on the interaction of plants and nanoparticles are laboratory or green-house studies, and they mainly target early plant growth development under regulated conditions (Prasad et al. 2012, Kořenková et al. 2017, Singh et al. 2019, Wagner et al. 2016). Consequently, there are only rare long-term field studies on NPs ability to accumulate elements required for best fruit quality.”.

Reviewer 2 Report

I acknowledge the effort made by the authors amending the manuscript that has noticeable improved. However, there are still several aspects that need attention.

There was a request to justify the use of sun flower. There is now a sentence justifying its commercial importance, but there is nothing mentioned on why has this species being used for the application of nanoparticles. Is this because of their broad leaves and rather big stomata? Is that because it is know its ability of absorb nutrients through the leaves, because they accumulate metals in several plants parts….why?

The authors claim that they have included scientific hypotheses to be tested. I cannot find them. If included, they should be presented in the introduction and they should be actual hypotheses and not mere predictions. On line 88 they use the word hypothesis, but actual scientific hypotheses are not presented. Please, revise and improve this end.

In the previous review there was a claim to explain why were those nutrients added, why was that the variety of sunflower used and the same for the fungicide and how it can interfere with the nanoparticle application. Similarly, it is important to explain why there were two applications of nanoparticle fertilization and why it was apply at the two different phonological stages.

Only the reason for the use of the sunflower variety is mentioned, unlikely the rest of the queries. It is important to explain how the treatments were applied, but the question is why did you do so? Why did you apply nutrients and fungicide when you did it?

In the previous review it was said ‘On line 176 it is said ‘The translocated and accumulated zinc and titanium in the sunflower seeds’. However, nothing is explained how translocation was estimated.’ This has not been amended. The answer provided does not help in understanding the current manuscript. I’s suggest the authors to be more specific for this particular methodology here.

I’m sorry to say that the discussion is almost in the same place where it was. Please, split results and discussion into two and provide to separate sections with a proper discussion. In its current form, this section is of little relevance.

The presented conclusions still need editing and be presented in the view of the hypotheses.

Author Response

Reviewer 2

  1. There was a request to justify the use of sun flower. There is now a sentence justifying its commercial importance, but there is nothing mentioned on why has this species being used for the application of nanoparticles. Is this because of their broad leaves and rather big stomata? Is that because it is know its ability of absorb nutrients through the leaves, because they accumulate metals in several plants parts….why?

#Answer:

We agree, this information were added to manuscript with new relevant references L.61-64: “…Research reports indicate that the sunflower disposes of broad leaves (Kaya 2016) and it has appropriate stomata morphology (Kirkham 2014) and also the ability to absorb, differentiate and accumulate metals in several parts of the plant. This is especially evident in the leaf structures (Li et al. 2019, Larue et al. 2014, Larue et al. 2012, Torabian et al. 2016, Kötschau et al. 2013).

New references:

Torabian, S., Zahedi, M. and Khoshgoftar, A.H. (2016) Effects of foliar spray of two kinds of zinc oxide on the growth and ion concentration of sunflower cultivars under salt stress. Journal of Plant Nutrition 39(2), 172-180.

Kötschau, A., A., Büchel, G., Einax, J.W., Fischer, C., von Tümpling, W. and Merten, D. (2013) Mapping of macro and micro elements in the leaves of sunflower (Helianthus annuus) by Laser Ablation–ICP–MS. Microchemical Journal 110, 783-789.

Kirkham, M.B. (2014) Principles of Soil and Plant Water Relations (Second Edition). Kirkham, M.B. (ed), pp. 409-430, Academic Press, Boston.

Kaya, Y. (2016) Breeding Oilseed Crops for Sustainable Production. Gupta, S.K. (ed), pp. 55-88, Academic Press, San Diego.

  1. The authors claim that they have included scientific hypotheses to be tested. I cannot find them. If included, they should be presented in the introduction and they should be actual hypotheses and not mere predictions. On line 88 they use the word hypothesis, but actual scientific hypotheses are not presented. Please, revise and improve this end.

#Answer:

We agree, hypothesis was included in manuscript L.94-100 “The sunflower has great ability to orientate against excessive sunlight radiation which could cause photocatalysis and accelerate both types of nanoparticles as photoactive nano-domains against selective photosynthetic plants. We predicted the various sunflower-responses based on the nanoparticles’ distinct photo-corrosive ability and subsequent assimilation by leaves with different bioavailability and functional metal pathways. This comparison includes the sunflower’s quantitative, nutritional and physiological parameters and its leaf morphological properties and fruit metal content.”.

  1. In the previous review there was a claim to explain why were those nutrients added, why was that the variety of sunflower used and the same for the fungicide and how it can interfere with the nanoparticle application. Similarly, it is important to explain why there were two applications of nanoparticle fertilization and why it was apply at the two different phonological stages.

#Answer:

We agree, applications of nanoparticle fertilization were added to manuscript with relevant references L.167-175. “Ensuring adequate nutrient supply before flower initiation increases the number of sunflower grains and root biomass. While late fertiliser application only partly modifies weight per grain and mainly affects plant protein concentration and decreases oil concentration, application in the early crop stages can stimulate lush biomass. However, it also affects leaf area index (LAI) after flowering by disease proliferation, or by excessive water consumption which limits normal provision during grain-filling in limited water input conditions (Alberio et al. 2015). Finally, common cultivation practice applies foliar fertilisers at the two sunflower growth stages 40 and 80 days after planting, and our methodology follows the foliar application for growth conducted by Ernst et al. (Ernst et al. 2016).

New used references:

Alberio, C., Izquierdo, N.G. and Aguirrezábal, L.A.N. (2015) Sunflower. Martínez-Force, E., Dunford, N.T. and Salas, J.J. (eds), pp. 53-91, AOCS Press.

Ernst, D., Kovar, M. and Černý, I. (2016) Effect of two different plant growth regulators on production traits of sunflower. Journal of Central European Agriculture 17(4), 998-1012.

  1. Only the reason for the use of the sunflower variety is mentioned, unlikely the rest of the queries. It is important to explain how the treatments were applied, but the question is why did you do so? Why did you apply nutrients and fungicide when you did it?

#Answer:

We agree, detailed information with appropriate reference was included in manuscript L.156-161: We must fully respect the principles of the sunflower cultivation system to achieve the required yield quantity and quality. This is based on the application of fertilisers, herbicides and fungicides (Kaya 2016, Alberio et al. 2015). Growth stimulator or foliar fertiliser application can be considered a cultivation system superstructure which has a positive effect on production, nutrition and physiological parameters in a changing climate (Koutroubas et al. 2014). These effects were confirmed in our previous sunflower experiments (Ernst et al. 2016).

New reference:

Koutroubas, S.D. – Vassiliou, G. – Damalas, C.A. 2014. Sunflower morphology and yield as affected by foliar applications of plant growth regulators. In International Journal of Plant Production, vol. 8, no. 2, pp. 215–230. ISSN 1735-6814. DOI: dx.doi.org/10.1590/S0100-83582015000100015

  1. In the previous review it was said ‘On line 176 it is said ‘The translocated and accumulated zinc and titanium in the sunflower seeds’. However, nothing is explained how translocation was estimated.’ This has not been amended. The answer provided does not help in understanding the current manuscript. I’s suggest the authors to be more specific for this particular methodology here.

#Answer:

We agree, manuscript were re-write according to suggestion from reviewer L. 200-207“The translocated and accumulated zinc and titanium in the kernels and hulls of fully ripe sunflower seeds were analysed by ICP-MS (Thermo Scientific iCap-Q, Germany). We then followed the standard measurement preparation; 0.15-0.30 g seed samples were digested in PTFE pressure vessels in the Anton Paar Multiwave 3000 microwave with concentrated 4 mL HNO3 and 2 mL H2O2 at 60 barometric pressure. The total zinc and titanium concentrations were determined by ICP-MS in KED mode with helium gas, and the calibration solutions were prepared from MERCK CertiPUR ICP 1000 mg.L-1 single-element standard solutions (Germany). Scandium and rhodium provided the internal standards.”

  1. I’m sorry to say that the discussion is almost in the same place where it was. Please, split results and discussion into two and provide to separate sections with a proper discussion. In its current form, this section is of little relevance.

#Answer:

Suggested separation to individual sections “Results”, and “Discussion” neither destroy this section nor completely disorganize whole concept of manuscript. Therefore, we decide to keep original manuscript framework, however, there were accepted many changes from reviewers’ comments before (shown blue watermarks, also with radical English corrections) including R+D section.

Of course, we fully agree to accept really constructive (before mentioned) comments, and suggestions (what has currently yellow watermarks) such as:

  • i) the explanation on why this species benefits from the application of the nanoparticles, L. 61-64“Research reports indicate that the sunflower disposes of broad leaves (Kaya 2016) and it has appropriate stomata morphology (Kirkham 2014) and also the ability to absorb, differentiate and accumulate metals in several parts of the plant. This is especially evident in the leaf structures”,
  • - or L. “298 – 317 Metal oxides such as ZnO, TiO2, CuO and Al2O3 are used in nano-fertilisers to boost crop growth (Sabir et al. 2014), and ZnO nano-fertilisers particularly provide an alternative to conventional chemical fertilisers by introducing the micro-nutrients required for efficient plant growth and development (Ditta and Arshad 2016, Monreal et al. 2016, Priyanka et al. 2019). General zinc uses include its benefits in catalytic activity; such as its function in dehydrogenases, aldolases, isomerases, transphosphorylases and RNA and DNA polymerases. Zinc is also important in tryptophan synthesis, cell division and maintaining membrane structure and potential, and it is beneficial in both photosynthesis and as a regulatory cofactor in protein synthesis (Chaudhuri and Malodia 2017). Moreover, zinc has previously been recorded herein as an essential micronutrient for plant growth and development (Tarafdar et al. 2014), and our research into its application to the Neostar hybrid sunflower cultivar recorded greater resistance to both drought and water-logging. Zinc therefore has a most important function in our hybrid’s production process because the ZnO nanoparticles are quickly transported into the sunflower and participate in its metabolic processes (Jabeen et al. 2017). TiO2, SiO2, and carbon nanotubes are also part of the new generation of nanoparticle fertilisers. While these have promoted plant growth, controversy surrounds their use because of their potential toxicity. However, beneficial results have been reported by Lu et al. (Lu et al. 2002) who recorded increased nitrogen fixation in Glycine maximum and improved seed germination and growth using a TiO2 and SiO2 mixture and Gao et al. (Gao et al. 2006) demonstrated that TiO2 alone increased total nitrogen, protein and chlorophyll content in the Spinacia oleracea species. Our results revealed no statistically significant difference in the number of plants and heads between the plants sprayed with ZnO-NPs and the NP-free controls.”
  • ii) or integration of all the results and how they together explain the behaviour of the plants L. 248-258 “The sunflower’s effective nutrient uptake, development, growth and metabolic functions depend on a combination of factors. This is common to all plants undergoing improvement by NPs foliar application, and the most important factors include the plant species and cultivar, ambient light and water conditions and the NPs particle size, stoichiometry, crystallinity and concentration (Nair et al. 2010, Wang et al. 2013). The two most usual nanoparticle penetration sites are cuticular and stomatal (Eichert et al. 2008), but only the stomatal pathway was available to our research Larue, et al. (Larue et al. 2014) because the cuticular pathway requires less than 5 nm NP-size and we were limited to 10nm-1μm NP-size with relatively high transport velocity (Eichert et al. 2008). More precisely, both our nanoparticle types are less than 30-nm, and they have good stoichiometry and high crystallinity with typical morphology (Tab.3, Fig.3). However, because of their different metal base, we expected significantly different photo-chemical behaviour under the plant’s sunlight radiation, specialised leaf anatomy and over-all plant-response.
  • iii) or we evaluated NPs photo-catalytical effect as main “driving force” which hypothetically are beyond the scope of manuscript L.94-98” Additionally, sunflower has predominant ability to orientate against sunlight radiation which could photocatalyticaly accelerate both types of nanoparticles as photoactive nano-domain against selective photosynthesis plant. Based on distinctive photo-corrosion ability of nanoparticles, and subsequent assimilation to leaves with different bioavailability, and metals-function pathways, we predicted the various sunflower-responses take place.”, L.370-379 ”However, chlorophyll content increases when the sunflower has foliar application of suitable concentrations of crystalline anastase TiO2-NPs in adequate environmental conditions. These conditions include sunlight, higher temperature and appropriate precipitation, and the enhanced chlorophyll content then enables plants to synthesise more light-harvesting pigment-protein complexes (LHCII) which absorb greater light energy. Modification was employed because nano-size anatase encourages oxidation-reduction under specific light wavelengths, and this enhances charge transfer between the nanoparticles and LHCII with resultant photosynthesis increase (Kuang 2003). In support, Lei et al. (Lei et al. 2007) identified that TiO2-NPs under visible and ultra-violet radiation significantly enhances the entire chlorophyll chain electron transport, photosystem II reduction, oxygen production and photo-phosphorylation.”
  • iv) or contribution to titanium vs soil system, L. 441-443 “Finally, Lyu et al. (Lyu et al. 2017) consider that low titanium soil concentration is beneficial for plant production because it simulates certain enzyme enhancement of essential nutrient uptake like Fe, increase crop yield and promote decrease in stress tolerance.”
  • v) in results &discussion section we include more than 10 new relevant references, and for all body of manuscript more than 20 citations.

New references:

Tarafdar, J.C., Raliya, R., Mahawar, H., Rathore, I., 2014. Development of zinc nanofertilizer to enhance crop produc-tion in pearl millet (Pennisetum americanum). Agribiol. Res. 3, 257–262.

Chaudhuri, S.K., Malodia, L., 2017. Biosynthesis of zinc oxide nanoparticles using leaf extract of Calotropis gigantea: characterization and its evaluation on tree seedling growth in nursery stage. Appl. Nanosci. 7, 501–512.

Jabeen, N., Maqbool, Q., Bibi, T., Nazar, M., Hussain, S.Z., Hussain, T., Anwaar, S., 2017. Optimised synthesis of ZnO-nano-fertiliser through green chemistry: boosted growth dynamics of economically important L. esculentum. IET Nanobiotechnol. 12, 405–411.

Ditta, A., Arshad, M., 2016. Applications and perspectives of using nanomaterials for sustainable plant nutrition. Nanotechnol. Rev. 5, 209–229. The presented conclusions still need editing and be presented in the view of the hypotheses.

Monreal, C.M., DeRosa, M., Mallubhotla, S.C., Bindraban, P.S., Dimkpa, C., 2016. Nanotechnologies for increasing the crop use efficiency of fertilizer-micronutrients. Biol. Fertil. Soils 52, 423–437.

Lu, C., Zhang, C., Wen, J., Wu, G., Tao, M., 2002. Research of the effect of nanometer materials on germination and growth enhancement of Glycine max and its mechanism. Soybean Sci. 21, 168–171.

Gao, F., Hong, F., Liu, C., Zheng, L., Su, M., Wu, X., Yang, P., 2006. Mechanism of nano-anatase TiO 2 on promoting photosynthetic carbon reaction of spinach. Biol. Trace Elem. Res. 111, 239–253.

Khodakovskaya, M.V., De Silva, K., Biris, A.S., Dervishi, E., Villagarcia, H., 2012. Carbon nanotubes induce growth enhancement of tobacco cells. ACS Nano 6, 2128–2135.

Sabir, S., Arshad, M., Chaudhari, S.K., 2014. Zinc oxide nanoparticles for revolutionizing agriculture: synthesis and applications. Sci. World J. 2014, 8.

Srinivasan, C., Saraswathi, R., 2010. Nano-agriculture—carbon nanotubes enhance tomato seed germination and plant growth. Curr. Sci. 99, 274–275.

Priyanka, N., et al. “Role of Engineered Zinc and Copper Oxide Nanoparticles in Promoting Plant Growth and Yield: Present Status and Future Prospects.” Advances in Phytonanotechnology, 2019, pp. 183–201., doi:10.1016/b978-0-12-815322-2.00007-9.

  1. The presented conclusions still need editing and be presented in the view of the hypotheses.

#Answer:

We were completely rewrite conclusion according to suggestion L. 455-479 “Our hypothesis is based on the sunflower’s extensive ability as a photo-sensitive plant to combat sunlight radiation, and here the foliar application of two types of photo-active nano-fertilisers accelerated both predictable and unpredictable plant functions and reactions.

We discovered that the TiO2-NPs treated variant improved sunflower quantitative and nutritional parameters including oil content. The TiO2-NPs provided unexpected early plant maturation with all dependent physiological indices. These results were most surprising, because although TiO2-NP plant enhancers are more photo-stable, they provide less nutrition and are potentially toxic. In contrast, the photo-corrosive and less resistant ZnO-NPs treated variant predicted, and better reflected, the sunflower physiological parameters with relatively good quantitative and nutritional effect. The most likely reason for its superiority is the increased zinc bioavailability following ZnO-NP transformation, and this supports both the sunflower physiology and its metabolic pathways.

There is no doubt that sunflower foliar application of low nanoparticle concentrations far surpasses NP-free control’s agronomic, environmental and economic benefits. This is clearly evident in the comparison of most plant parameters. There are also significant differences in trichome distribution between the NPs-treatment methods and the control, and this is especially noted in trichome width and length on the surface of leaves collected at the flower-bud development stage. Moreover, neither nanoparticle treatment detrimentally affected final food quality. This was proven by analysis which showed no higher Zn or Ti translocation to fully ripe sunflower seeds compared to the NPs-free control.

In conclusion, although nano-fertilisers are widely used to enhance plant production, yield and fruit quality, their effects remain unpredictable. There are also unexplored issues. These include the NPs fertiliser mode of action and uptake mechanisms and dose-dependent plant response. Agricultural nanotechnology therefore remains one of the most challenging disciplines in combatting current climate change.”